# Genetic diversity and phylogeographic patterns of the peacock jewel-damselfly, *Rhinocypha fenestrella* (Rambur, 1842)

**Mamat Noorhidayah[1], Noor Azrizal-Wahid[2]\*, Van Lun Low[3], Norma-Rashid Yusoff[1]**

**1** Institute of Biological Sciences, Faculty of Science, Universiti Malaya, Kuala Lumpur, Malaysia,
**2** Department of Biology, Faculty of Science, Universiti Putra Malaysia, Serdang, Selangor, Malaysia,
**3** Higher Institution Centre of Excellence (HICoE), Tropical Infectious Diseases Research & Education Centre (TIDREC), Universiti Malaya, Kuala Lumpur, Malaysia

\* noorazrizal@upm.edu.my

**Data Availability Statement:** All relevant data are within the manuscript and its Supporting information files.

## Abstract

Despite is known to have widespread distribution and the most active species of the family Chlorocyphidae, the molecular data of *Rhinocypha fenestrella* (Rambur, 1842) are relatively scarce. The present study is the first that examined the genetic diversity and phylogeographic pattern of the peacock jewel-damselfly *R. fenestrella* by sequencing the cytochrome C oxidase I (*cox1*) and 16S rRNA gene regions from 147 individuals representing eight populations in Malaysia. A total of 26 and 10 unique haplotypes were revealed by the *cox1* and 16S rRNA genes, respectively, and 32 haplotypes were recovered by the concatenated sequences of *cox1*+16S. Analyses indicated that haplotype AB2 was the most frequent and the most widespread haplotype in Malaysia while haplotype AB1 was suggested as the common ancestor haplotype of the *R. fenestrella* that may arose from the Negeri Sembilan as discovered from *cox1*+16S haplotype network analysis. Overall haplotype and nucleotide diversities of the concatenated sequences were $H_d = 0.8937$ and $P_i = 0.0028$, respectively, with great genetic differentiation ($F_{ST} = 0.6387$) and low gene flow ($N_m = 0.14$). Population from Pahang presented the highest genetic diversity ($H_d = 0.8889$, $P_i = 0.0022$, $N_h = 9$), whereas Kedah population demonstrated the lowest diversity ($H_d = 0.2842$, $P_i = 0.0003$, $N_h = 4$). The concatenated sequences of *cox1*+16S showed genetic divergence ranging from 0.09% to 0.97%, whereas the genetic divergence for *cox1* and 16S rRNA genes were 0.16% to 1.63% and 0.01% to 0.75% respectively. This study provides for the first-time insights on the intraspecific genetic diversity, phylogeographic pattern and ancestral haplotype of *Rhinocypha fenestrella*. The understanding of molecular data especially phylogeographic pattern can enhance the knowledge about insect origin, their diversity, and capability to disperse in particular environments.

## Introduction

*Rhinocypha* (Cowley, 1937), from the family of Chlorocyphidae, is the most characteristic genus of damselflies in tropical Asia. Rhinocyphae is of a very particular nature in their habitat

**Funding:** Author who received the funding award= Norma-Rashid Yusoff; Grant numbers awarded to the author= 1; The full name of funder= The Universiti Malaya Research Grant (UMRG) PG065 – 2015 and ST061-2022; URL of each funder website= https://umhvl.um.edu.my/grants-and-projects-awarded; Role of funder: Study design, conceptualization, data collection and analysis, preparation of the manuscript draft.

**Competing interests:** The authors have declared that no competing interests exist.

[1] and certain species can adapt to and tolerate disturbed habitats [2]. This characteristic makes them good "thermometers" of environment quality, thus good study subjects for phylogeography [3]. *Rhinocypha fenestrella* (Rambur, 1842), syn, *Aristocypha fenestrella*, also known as a peacock jewel, is the most widespread species in the genus, and one of the most active species of the family, which occurs in Peninsular Malaysia, Thailand, Burma, Laos, Vietnam, and southern China [4]. It is one of several common mountain stream damselflies and is usually found in the primary forests [1].

Malaysia is known to be one of the three mega-biodiversity countries in Southeast Asia, however, the phylogeographic pattern of damselflies has been relatively scarce, especially for this particular species. Several phylogeographic studies have been published concerning odonates [5–7], however, until now, no comparable research related to odonate species in Malaysia has been conducted. Instead, the species with a widespread distribution that includes dragonflies normally will have complexes of multiple lineages or variations in the genetic diversity with the geographic region [8–12]. From a point of zoogeography, the Rhinocyphae are of substantial importance, for instance, in the Malay Archipelago, each large island, mostly has its own group of endemic species [1].

A phylogeographic analysis is a powerful method of obtaining insights into the historical processes that have shaped the species' temporal distribution and genetic variation [13]. Recent phylogeographic research on widely distributed odonatan species has indicated different responses to historical climatic changes in space and time [14,15]. With the advances in molecular techniques, mitochondrial DNA has been identified as an excellent genetic marker of gene flow in matrilineal inheritance [16] and it is the most widely used marker to study the molecular ecology in animal taxa [17,18].

Particularly, cytochrome C oxidase subunit I (*cox1*) and 16S ribosomal RNA (16S) are known to be reliable genetic markers and the commonly applied markers in Odonata [7,19–21]. In addition to nuclear markers, mitochondrial markers were also provided well-resolved and supported trees from species to family level [22,23]. Given the high resolution of mitochondria-encoded *cox1* and 16S genes reported in odonates [24], this study for the first time attempts to characterize the genetic diversity and population structure of *R. fenestrella*, across its range in Malaysia.

## Materials and methods

### Sample collection

A total of 147 *Rhinocypha fenestrella* individuals were collected from eight populations representing eight states in Peninsular Malaysia from the period of 2014 to 2015 with the permission of the Forestry Department Peninsular Malaysia (Permit Number: JH/100 Jld.7 (12)) (Fig 1 and Table 1). The identification of *R. fenestrella* was performed according to morphological descriptions and taxonomic keys as described by Orr and Hamalainen (2003) [25], and through personal experience. Methods of sampling and preservation of Odonata were based on previously described standard methods [26]. Generally, samples were caught by using a sweep net and were dried preserved for morphological identification, while the legs were removed from each individual and stored in a vial containing 80% ethanol for molecular works.

### DNA extraction and amplification

Genomic DNA was extracted from four to six legs of each ethanol-preserved specimen using the i-genomic CTB DNA Extraction Mini Kit (iNtRON Biotechnology Inc., Seongnam, South Korea). The DNA amplifications of both *cox1* and 16S genes were conducted using an Applied

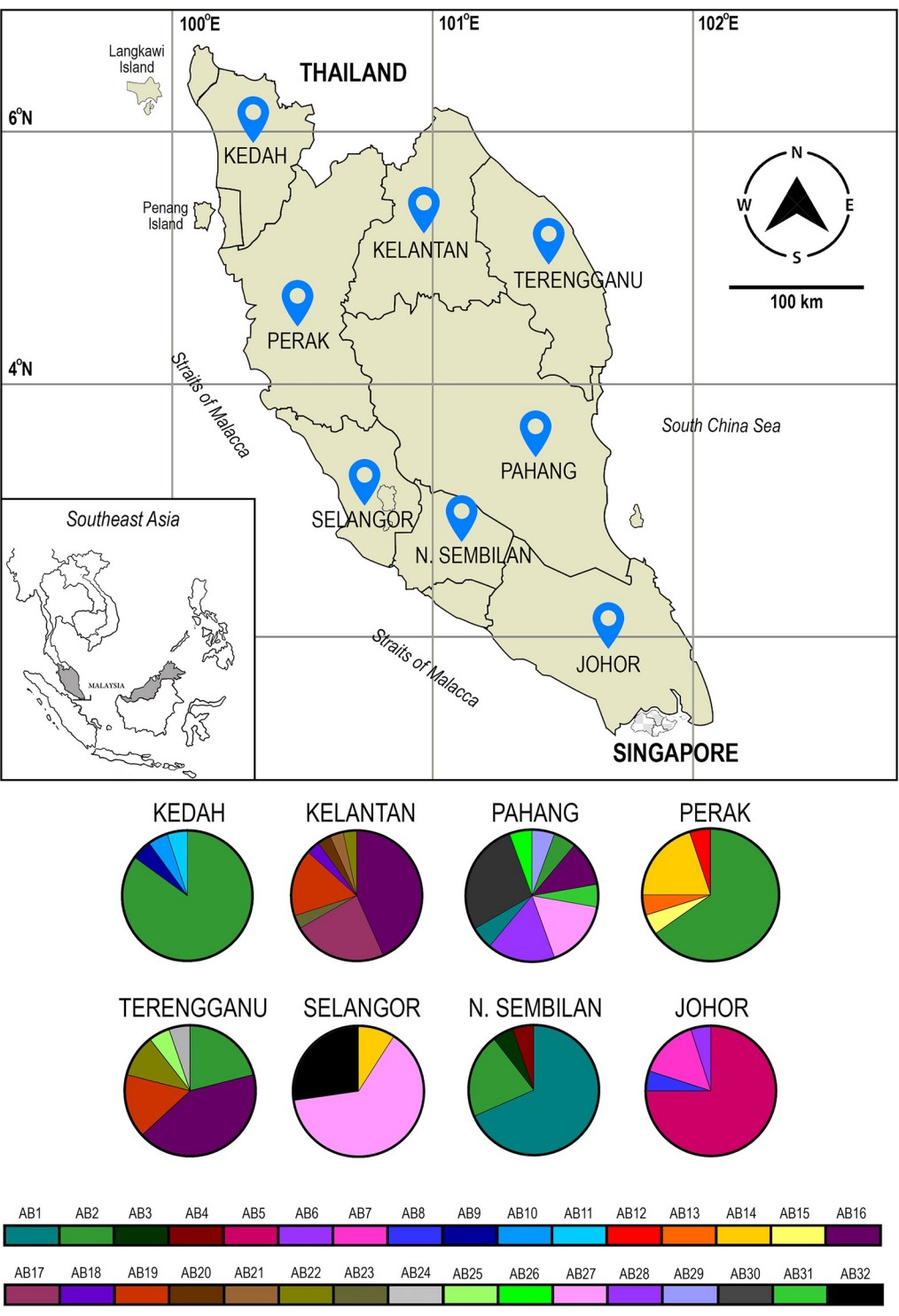

**Fig 1. Map of sampling sites and haplotype distribution (AB1–AB15) of concatenated *cox1*+16S sequences for *Rhinocypha fenestrella* from eight sampling locations representing eight states of Peninsular Malaysia.**

**Table 1. Sampling localities and geographic position of sampling sites of *Rhinocypha fenestrella*.**

| State | District | Specific Locality | Geographic position |
|---|---|---|---|
| N. Sembilan | Jelebu | Jeram Toi Waterfall | N 02 51' 52.7" E 102 00' 52.0" |
| Johor | Bekok | Sungai Bantang Waterfall | N 02 20' 46.7" E 103 09' 23.9" |
| Kedah | Baling | Lata Bayu Waterfall | N 05 43' 02.9" E 100 48' 50.9" |
| Perak | Tapah | Lata Iskandar Waterfall | N 04 19' 27.18" E 101 19' 31.36" |
| Kelantan | Machang | Jeram Linang Waterfall | N 05 44' 33.4" E 102 22' 26.2" |
| Terengganu | Kuala Berang | Sekayu Waterfall | N 04 57' 45.8" E 102 57' 11.9" |
| Pahang | Bentong | Chamang Waterfall | N 03 30' 34.2" E 101 51' 36.3" |
| Selangor | Hulu Selangor | Sungai Sendat Waterfall | N 03 24' 14.4" E 101 41' 00.5" |

Biosystems Veriti 96-Well Thermal Cycler (Applied Biosystems Inc., Foster City, CA, USA) with the amplification protocol consisting of 30 sec at 94°C followed by 35 cycles of 50 sec at 94°C, 50 sec at 50°C and 50 sec at 72°C, and a final 7 min at 72°C. Primers for amplification of the *cox1* gene were 5'- GGT CAA CAA ATC ATA AAG ATA TTG G - 3' for forward primer [27] and 5'- GGA TGG CCA AAA AAT CAA AAT AAA TG -3' for reverse primer [28]. For the 16S gene, ODO 12852 and ODO 13393 primer sets (forward primer, 5'- AGA AAC CGA CCT GGC TTA AA -3'; reverse primer, 5'- CGC CTG TTT ATC AAA AAC AT -3') were utilized [7]. Each PCR amplification was performed in a reaction mixture containing 50–100 ng of genomic DNA, 25 µl of NEXpro e-PCR 2X Master Mix (Genes Labs Inc., Gyeonggi-do, South Korea), and 10 pmol of each forward and reverse primer.

## DNA purification, sequencing, and alignment

The amplified samples were then electrophoresed on 2% agarose gel pre-stained with SYBR Safe™ (Invitrogen Corp., Carlsbad, CA, U.S.A.), and the PCR products were sent outsource to a commercial company (Apical Scientific Sdn. Bhd. Selangor, Malaysia) for DNA sequencing in both forward and reverse directions. The samples were sequenced using the BigDyeH Terminator 3.1 Sequencing Kit.

All sequences were assembled and edited using Molecular Evolutionary Genetics Analysis (MEGA) software Version 11.0 [29] and BioEdit 7.2 [30] and preliminarily aligned using CLUSTALX [31]. The aligned sequences were then subjected to a BLAST search (www.ncbi.nlm.nih.gov/blast/) for species validation.

## Genetic diversity and haplotype analyses

The aligned *cox1* and 16S sequences at first were analysed separately, then were concatenated to yield a total length for further analysis. Molecular characteristics such as the number of haplotypes ($N_h$), haplotype diversity ($H_d$), nucleotide diversity ($P_i$), the number of segregating sites ($S$), and average number of sequence differences ($K$) were determined with the program DNASP® (DNA Sequence Polymorphism) v6.12.03 [32]. DNASP® was also used to perform neutrality tests including Tajima's $D$ [33] and Fu's $F_s$ [34].

The genetic differentiation ($F_{ST}$) and gene flow ($N_m$) pairwise among the *R. fenestrella* populations were determined using DNASP®. Whereas the significant level of $F_{ST}$ was determined using ARLEQUIN v3.5 [35]. The levels of genetic differentiation are defined as $F_{ST} > 0.25$ (great differentiation), $0.15 < F_{ST} < 0.25$ (moderate differentiation), and $F_{ST} < 0.15$ (negligible differentiation) according to the classification criteria by Wright (1978) [36]. The levels of gene flow are categorized as $N_m > 1$ (high gene flow), $0.25 < N_m < 0.99$ (intermediate gene

flow), and $N_m < 0.25$ (low gene flow) [37]. Molecular variance analysis (AMOVA) between populations was performed using ARLEQUIN v3.5 with 1000 permutations. Uncorrected pairwise distances (*p*-distance) were assessed using PAUP 4.0B10 to measure the genetic divergence of *R. fenestrella*. Moreover, the observed and expected distributions of the number of pairwise genetic differences (mismatch distributions) were performed using DNASP®.

Furthermore, to visualize the phylogeographic pattern between the populations, and to calculate the minimum number of mutational steps between the sequences, haplotype networks were constructed using TCS 1.13® [38] with a 95% parsimony criterion for both *cox1* and 16S respectively, and concatenated *cox1*+16S genes sequences.

## Results

### Genetic diversity of *Rhinocypha fenestrella*

The final lengths of aligned sequence fragment were 614 bp, 534 bp, and 1148 bp for *cox1*, 16S, and concatenated *cox1*+16S, respectively. The generated sequences that exhibit unique haplotypes in this study were deposited in the National Center for Biotechnology Information (NCBI) GenBank database under accession numbers KY678719–KY678744 for *cox1*, and KY678745–KY678754 for 16S genes.

The haplotype diversity ($H_d$) in a population for the *cox1* gene ranged from 0.1947 in the Kedah population to 0.8889 in the Pahang population. Nonetheless, for the 16S gene, it ranged from 0.0000 found in three populations (Johor, Perak, and Selangor) to 0.2842 in the Kelantan population. In concatenated *cox1*+16S sequence analyses, the overall value of haplotype diversity was 0.8937, and nucleotide diversity ($P_i$) was 0.0028. Kedah population showed the lowest diversity for both nucleotide and haplotype diversities ($H_d$ = 0.2842, $P_i$ = 0.0003), while the highest haplotype diversity was shown by Pahang ($H_d$ = 0.8889), and the highest nucleotide diversity was presented by the Kelantan population ($P_i$ = 0.0023) (Table 2). In total average data estimates, the *cox1* gene revealed higher for both haplotype and nucleotide diversities ($H_d$ = 0.8846, $P_i$ = 0.0051) than did the 16S gene ($H_d$ = 0.1292, $P_i$ = 0.0013).

The results of AMOVA showed that the genetic variation among the *R. fenestrella* populations accounted for 64.33% of the total variation, which was higher than the 35.67% of genetic variation observed within the populations, indicating that the genetic variation in *R. fenestrella* mainly occurred among populations (Table 3). The genetic divergence (0.6387), which was measured by the fixation index (FST) showed a great degree of genetic differentiation among *R. fenestrella* samples in Malaysia.

### Haplotype variation and distribution

A set of aligned sequences of 147 taxa of the *cox1* revealed 26 haplotypes (A1 –A26). For *cox1*, the most prevalent haplotype was A2 (*n* = 40) and was found in five populations. The second most-frequent haplotypes were A1, A4, and A14 (*n* = 15) while the least prevalent haplotypes presented as singleton were A3, A5, A7 –A11, A18 –A20, A23, and A25. Notably, based on the star-like pattern of the haplotype network (Fig 2), the haplotypes A2 –A26 from the *cox1* gene were considered to be originated from haplotype A1.

For the 16S gene, the sequences revealed 10 haplotypes (B1 –B10). The most prevalent haplotype was B1 (*n* = 140) while all other haplotypes except B8 (*n* = 2), appeared as singleton haplotypes. Haplotype B1 was found in all populations. The haplotype network of 16S gene showed a star-like pattern suggesting haplotype B1 as a common ancestor of all the others (Fig 3). Nevertheless, a median-joining network showed no obvious geographical pattern in haplotype distribution.

**Table 2. Genetic diversity indices and neutrality test based on *cox1*, 16S, and concatenated *cox1*+16S sequences of *Rhinocypha fenestrella* from eight different populations in Peninsular Malaysia.**

| Markers | n | Nh | Hd | Pi | S | K | D | Fs |
|---|---|---|---|---|---|---|---|---|
| **16S rRNA** | | | | | | | | |
| Johor | 20 | 1 | 0.0000 | 0.0000 | 0 | 0 | np | np |
| Kedah | 20 | 2 | 0.1000 | 0.0002 | 1 | 0.1000 | -1.1644 | -0.879 |
| Kelantan | 20 | 4 | 0.2842 | 0.0010 | 4 | 0.4947 | -1.9857* | -1.589 |
| N. Sembilan | 19 | 2 | 0.1053 | 0.0002 | 1 | 0.1053 | -1.1648 | -0.838 |
| Pahang | 20 | 3 | 0.1947 | 0.0054 | 15 | 2.7526 | -1.2996 | 5.371 |
| Perak | 20 | 1 | 0.0000 | 0.0000 | 0 | 0.0000 | np | np |
| Selangor | 11 | 1 | 0.0000 | 0.0000 | 0 | 0.0000 | np | np |
| Terengganu | 20 | 4 | 0.2842 | 0.0031 | 15 | 1.5947 | -2.4439* | 1.627 |
| **OVERALL** | **150** | **10** | **0.1292** | **0.0013** | **22** | **0.6850** | **-2.3808** | **-5.488** |
| ***cox1*** | | | | | | | | |
| Johor | 20 | 4 | 0.4316 | 0.001 | 3 | 0.4684 | -1.1914 | -1.713 |
| Kedah | 20 | 3 | 0.1947 | 0.000 | 2 | 0.2000 | -1.5128 | -1.863 |
| Kelantan | 20 | 4 | 0.7263 | 0.003 | 5 | 2.0158 | 1.3167 | 1.812 |
| N. Sembilan | 19 | 3 | 0.4328 | 0.0007 | 2 | 0.4561 | -0.4849 | -0.421 |
| Pahang | 18 | 9 | 0.8889 | 0.0040 | 9 | 2.4641 | -0.2066 | -2.838 |
| Perak | 20 | 5 | 0.5579 | 0.0013 | 5 | 0.7947 | -1.3344 | -1.711 |
| Selangor | 11 | 3 | 0.5636 | 0.0013 | 3 | 0.8000 | -0.7494 | 0.158 |
| Terengganu | 19 | 6 | 0.7778 | 0.0028 | 6 | 1.7193 | 0.0049 | -0.780 |
| **OVERALL** | **147** | **26** | **0.8846** | **0.0051** | **28** | **3.0981** | **-1.1123** | **-9.824** |
| ***cox1*+16S** | | | | | | | | |
| Johor | 20 | 4 | 0.4316 | 0.0004 | 3 | 0.4684 | -1.1914 | -1.713 |
| Kedah | 20 | 4 | 0.2842 | 0.0003 | 3 | 0.3000 | -1.7233 | -2.749 |
| Kelantan | 20 | 8 | 0.8211 | 0.0023 | 10 | 2.6105 | -0.5605 | -1.297 |
| N. Sembilan | 19 | 4 | 0.5088 | 0.0005 | 3 | 0.5614 | -0.9407 | -1.355 |
| Pahang | 18 | 9 | 0.8889 | 0.0021 | 9 | 2.4641 | -0.2066 | -2.838 |
| Perak | 20 | 5 | 0.5579 | 0.0007 | 5 | 0.7947 | -1.3344 | -1.711 |
| Selangor | 11 | 3 | 0.5636 | 0.0007 | 3 | 0.8000 | -0.7494 | 0.158 |
| Terengganu | 19 | 6 | 0.7778 | 0.0017 | 8 | 1.9298 | -0.5366 | -0.460 |
| **OVERALL** | **147** | **32** | **0.8937** | **0.0028** | **37** | **3.2341** | **-1.5732** | **-16.523** |

**Notes**: $n$ = number of sequences; $Nh$ = number of haplotypes; $Hd$ = haplotype diversity; $Pi$ = nucleotide diversity; $S$ = number of segregating sites; K = average number of sequence differences; $D$ = Tajima's; $F_s$ = Fu's.

*significant at p<0.05

**Table 3. Analysis of molecular variance (AMOVA) for *Rhinocypha fenestrella* collected from eight populations in Peninsular Malaysia.**

| Source of Variation | df | Sum of Squares | Variance Components | Percentage of Variance (%) |
|---|---|---|---|---|
| Among populations | 7 | 149.073 | 1.12884 Va | 64.33 |
| Within populations | 139 | 87.015 | 0.62601 Vb | 35.67 |
| Total | 146 | 236.088 | 1.75485 | |
| Fixation Index | FST | 0.63872 | | |

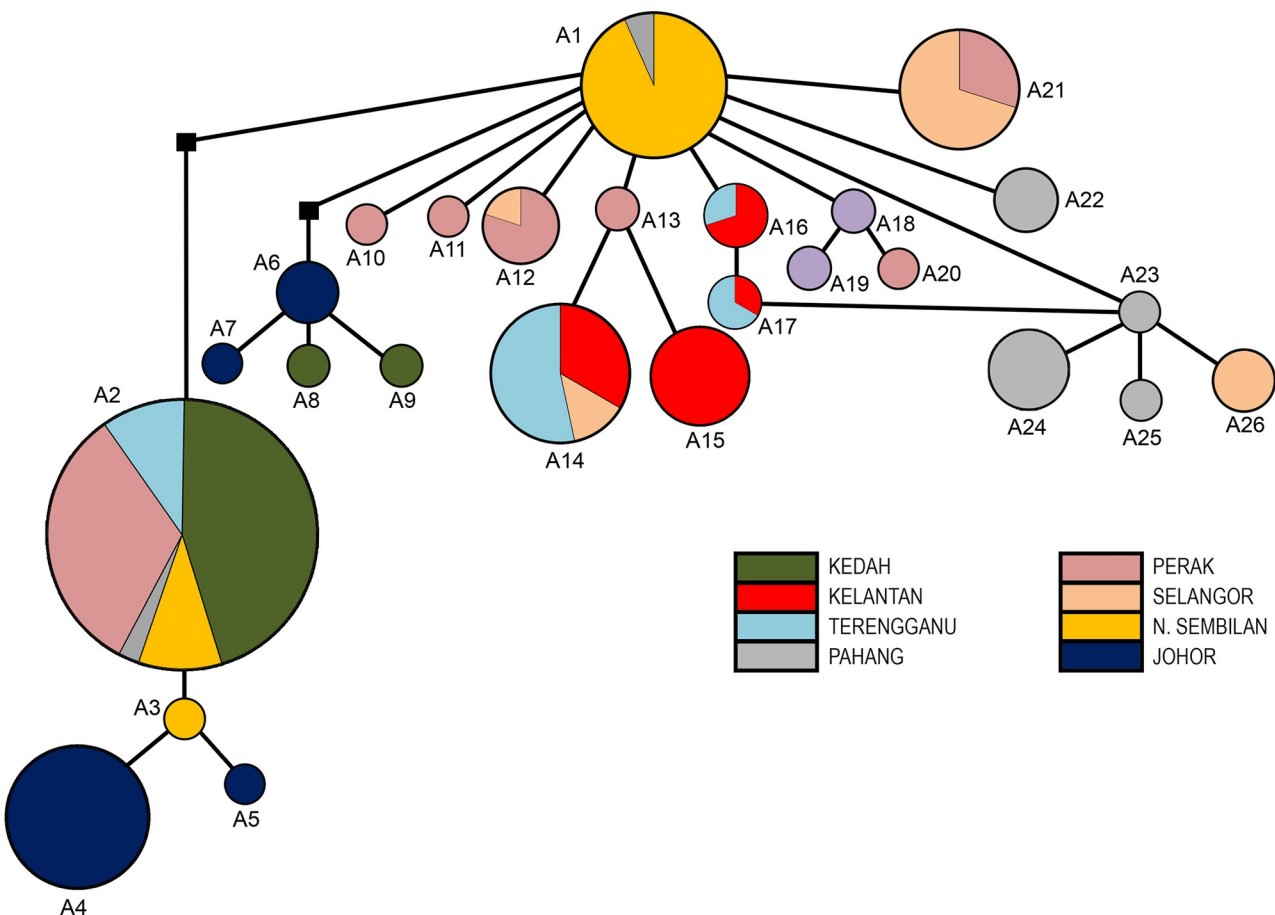

**Fig 2. Median joining haplotype network of *Rhinocypha fenestrella* of *cox1* sequences isolated from eight different states in Peninsular Malaysia.**
Each circle represents a haplotype variation, and the size of a circle is proportional to the number of sequences assigned to that haplotype. Circles of the same colour represent haplotypes from the same population. A small black square represents median vectors.

The concatenated *cox1*+16S sequences revealed 32 haplotypes (AB1 –AB32). Haplotype AB2 was the most frequent and the most widespread haplotype based on its prevalence in Malaysia, while there were 18 singleton haplotypes (AB3, AB4, AB6, AB8 –AB12, AB15, AB18, AB20, AB21, AB 23 –AB26, AB29, and AB31 (Table 4). Furthermore, the Kelantan population presented as the most diverse population that had the highest number of haplotypes ($N_h$ = 30) while the least number of haplotypes was presented in the Selangor population ($N_h$ = 11). Haplotype AB1 appeared to be as a common ancestor of the other haplotypes as indicated by the star-like pattern of concatenated *cox1*+16S haplotype network (Fig 4). Consequently, based of sampling sites, this study revealed that a recent common ancestor of the *R. fenestrella* species in Malaysia existed from the Negeri Sembilan population that constituted the most numbers of haplotype AB1.

## Haplotype genetic divergence

The *p*-distance between *R. fenestrella* haplotypes for *cox1* ranged from 0.16%– 1.63% with the highest value showed pairwise between haplotypes A5 with A15, A17, A19, and between haplotypes A7 with A15, A17, A19 (S1 Table). On the other hand, the genetic divergence of 16S

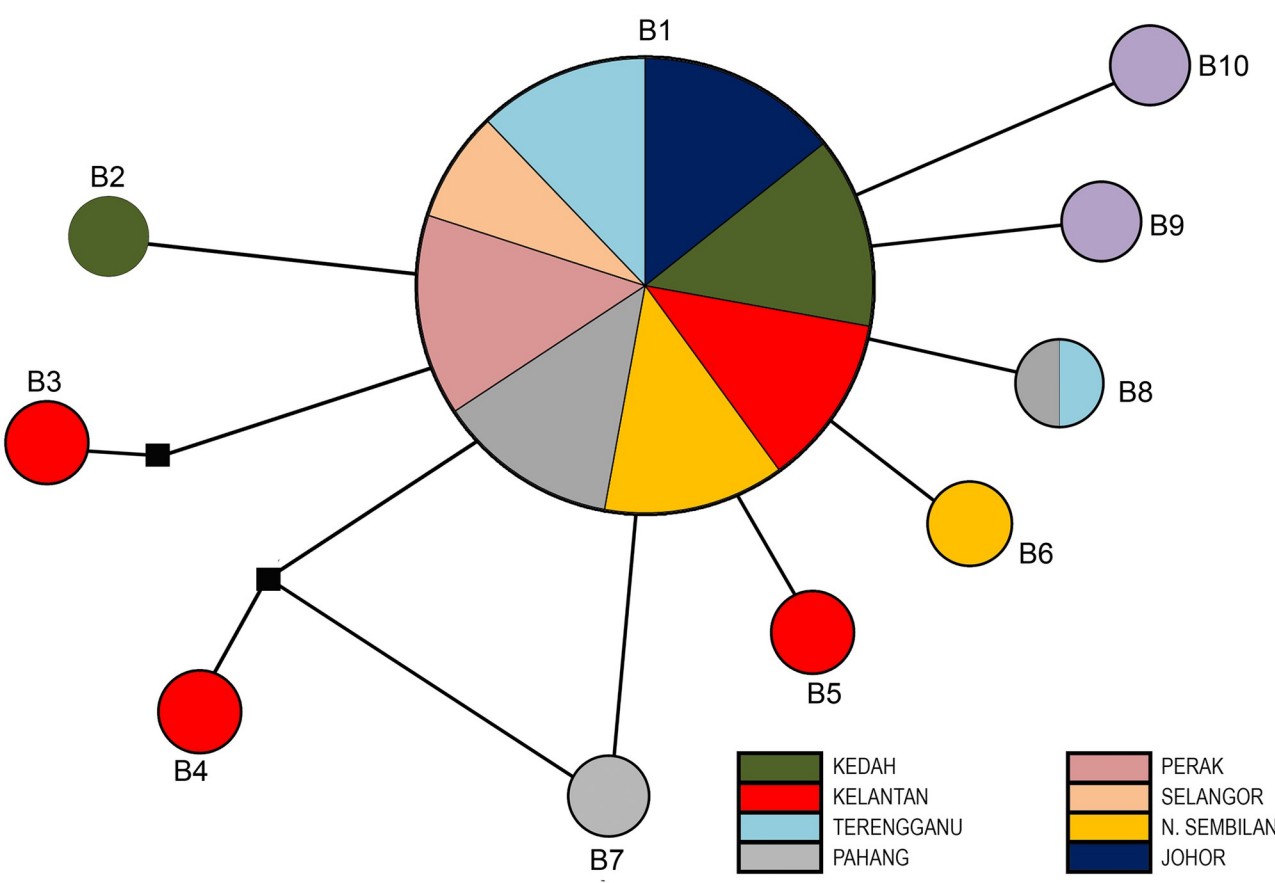

**Fig 3. Median joining haplotype network of *Rhinocypha fenestrella* of 16S sequences isolated from eight different states in Peninsular Malaysia.** Each circle represents a haplotype variation, and the size of a circle is proportional to the number of sequences assigned to that haplotype. Circles of the same colour represent haplotypes from the same population. A small black square represents median vectors.

haplotypes ranged from 0.01%– 0.75% with the highest value presented pairwise between haplotypes B5 and B7 (S2 Table). For the concatenated *cox1*+16S, the genetic divergence ranged from 0.09% to 0.97% (S3 Table).

In concatenated *cox1*+16S sequences, the neutrality test of Fu's $F_s$ showed negative values in all populations with an exception for Selangor. Additionally, Tajima's $D$ tests for all populations were indicated by negative values and were not statistically significant. The overall $F_{ST}$ is 0.6387 and $N_m$ is 0.14, suggesting that the genetic differentiation was great while there was a low gene flow among the populations of *R. fenestrella* in Malaysia. Based on Table 5, the highest $F_{ST}$ value (0.92494) appeared between Negeri Sembilan and Johor populations while the lowest value (0.03561) was between the populations of Kelantan and Terengganu. Moreover, in mismatch distribution analysis, the actual distribution curve showed a bimodal characteristic indicating no recent population expansion. Nevertheless, based on the graph pattern, historically, the population distribution undergone sharply increased, sudden decreased and then increased (Fig 5).

**Table 4. Haplotype distribution of *Rhinocypha fenestrella* (*n* = 147) from Peninsular Malaysia based on the concatenated *cox1*+16S sequences corresponding to the eight populations.**

| Population | n | Haplotype *cox1*+16S | | | | | | | | | | | | | | | | | | | | | | | | | | | | | | | |
|---|---|---|---|---|---|---|---|---|---|---|---|---|---|---|---|---|---|---|---|---|---|---|---|---|---|---|---|---|---|---|---|---|---|
| | | AB1 | AB2 | AB3 | AB4 | AB5 | AB6 | AB7 | AB8 | AB9 | AB10 | AB11 | AB12 | AB13 | AB14 | AB15 | AB16 | AB17 | AB18 | AB19 | AB20 | AB21 | AB22 | AB23 | AB24 | AB25 | AB26 | AB27 | AB28 | AB29 | AB30 | AB31 | AB32 |
| N. Sembilan | 19 | 13 | 4 | 1 | 1 | - | - | - | - | - | - | - | - | - | - | - | - | - | - | - | - | - | - | - | - | - | - | - | - | - | - | - | - |
| Johor | 20 | - | - | - | - | 15 | 1 | 3 | 1 | - | - | - | - | - | - | - | - | - | - | - | - | - | - | - | - | - | - | - | - | - | - | - | - |
| Kedah | 20 | - | 17 | - | - | - | - | - | - | 1 | 1 | 1 | - | - | - | - | - | - | - | - | - | - | - | - | - | - | - | - | - | - | - | - | - |
| Perak | 20 | - | 13 | - | - | - | - | - | - | - | - | - | 1 | 1 | 4 | 1 | - | - | - | - | - | - | - | - | - | - | - | - | - | - | - | - | - |
| Kelantan | 20 | - | - | - | - | - | - | - | - | - | - | - | - | - | - | - | 13 | 7 | - | - | - | - | - | - | - | - | - | - | - | - | - | - | - |
| Terengganu | 19 | - | 3 | - | - | - | - | - | - | - | - | - | - | - | - | - | 8 | - | 1 | 1 | 1 | 1 | 1 | 1 | 1 | 1 | - | - | - | - | - | - | - |
| Pahang | 18 | 1 | 1 | - | - | - | - | - | - | - | - | - | - | - | - | - | 2 | - | - | - | - | - | - | - | - | - | 1 | 3 | 3 | 1 | 5 | 1 | - |
| Selangor | 11 | - | - | - | - | - | - | - | - | - | - | - | - | 1 | - | - | - | - | - | - | - | - | - | - | - | - | - | 7 | - | - | - | - | 3 |
| TOTAL | 147 | 14 | 38 | 1 | 1 | 15 | 1 | 3 | 1 | 1 | 1 | 1 | 1 | 2 | 4 | 1 | 23 | 7 | 1 | 1 | 1 | 1 | 1 | 1 | 1 | 1 | 1 | 10 | 3 | 1 | 5 | 1 | 3 |

*n* = number of sequences

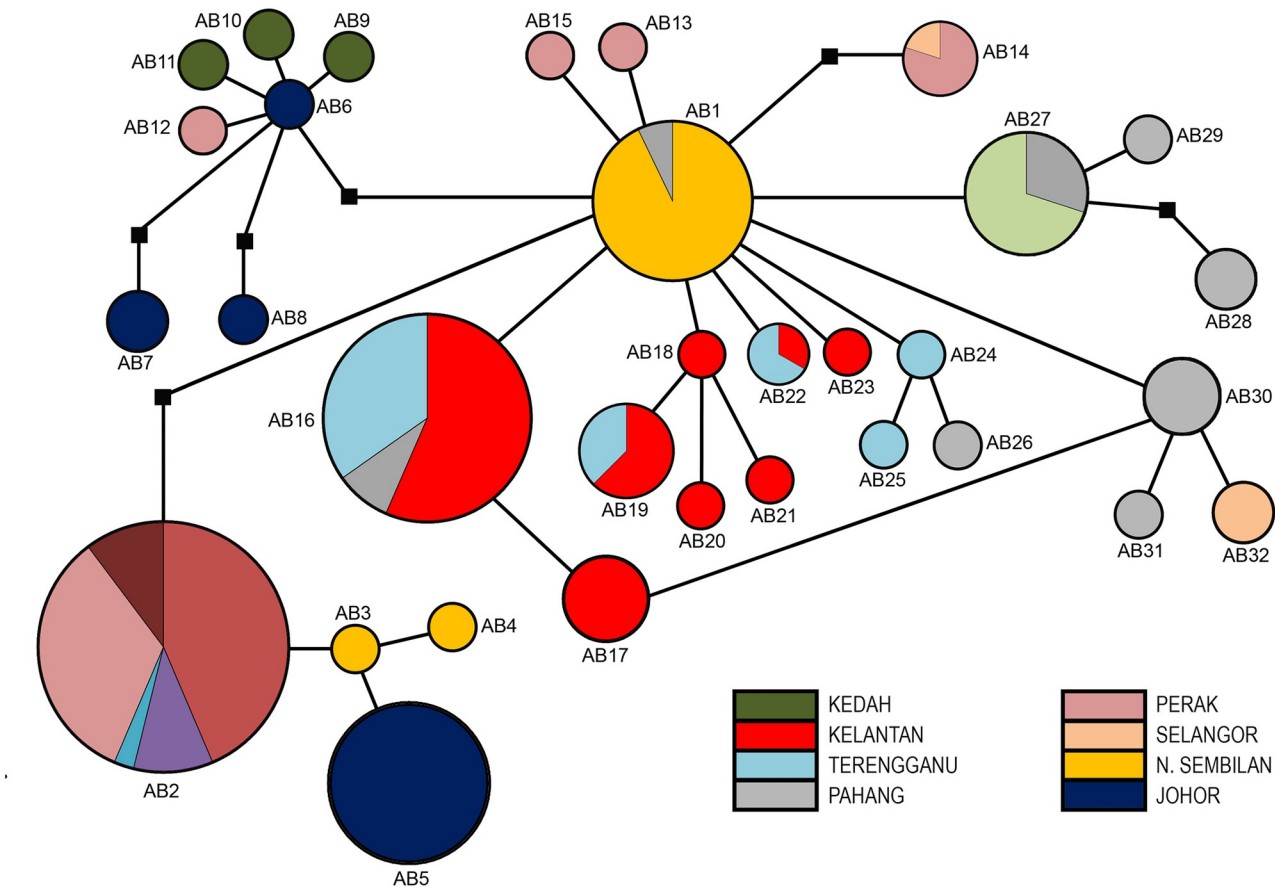

**Fig 4. Median joining haplotype network of *Rhinocypha fenestrella* of concatenated *cox1*+16S sequences isolated from eight different states in Peninsular Malaysia.** Each circle represents a haplotype variation, and the size of a circle is proportional to the number of sequences assigned to that haplotype. Circles of the same colour represent haplotypes from the same population. A small black square represents median vectors.

**Table 5. Pairwise genetic differentiation ($F_{ST}$) among *Rhinocypha fenestrella* in eight distinct populations of Peninsular Malaysia.**

| Population | S1 | S2 | S3 | S4 | S5 | S6 | S7 | S8 |
|---|---|---|---|---|---|---|---|---|
| [S1] N. Sembilan | | + | + | + | + | + | + | + |
| [S2] Johor | 0.92494 | | + | + | + | + | + | + |
| [S3] Kedah | 0.58987 | 0.93701 | | + | + | + | + | + |
| [S4] Perak | 0.49448 | 0.90132 | 0.08772 | | + | + | + | + |
| [S5] Kelantan | 0.50648 | 0.81452 | 0.41789 | 0.39192 | | − | + | + |
| [S6] Terengganu | 0.50694 | 0.84408 | 0.37966 | 0.34860 | 0.03561 | | + | + |
| [S7] Pahang | 0.32321 | 0.80936 | 0.16887 | 0.09112 | 0.28709 | 0.23649 | | + |
| [S8] Selangor | 0.69758 | 0.91882 | 0.65154 | 0.49765 | 0.48615 | 0.49847 | 0.23087 | |

$F_{ST}$ value on the below diagonal, and upper diagonal showing the significance (+, p<0.05)

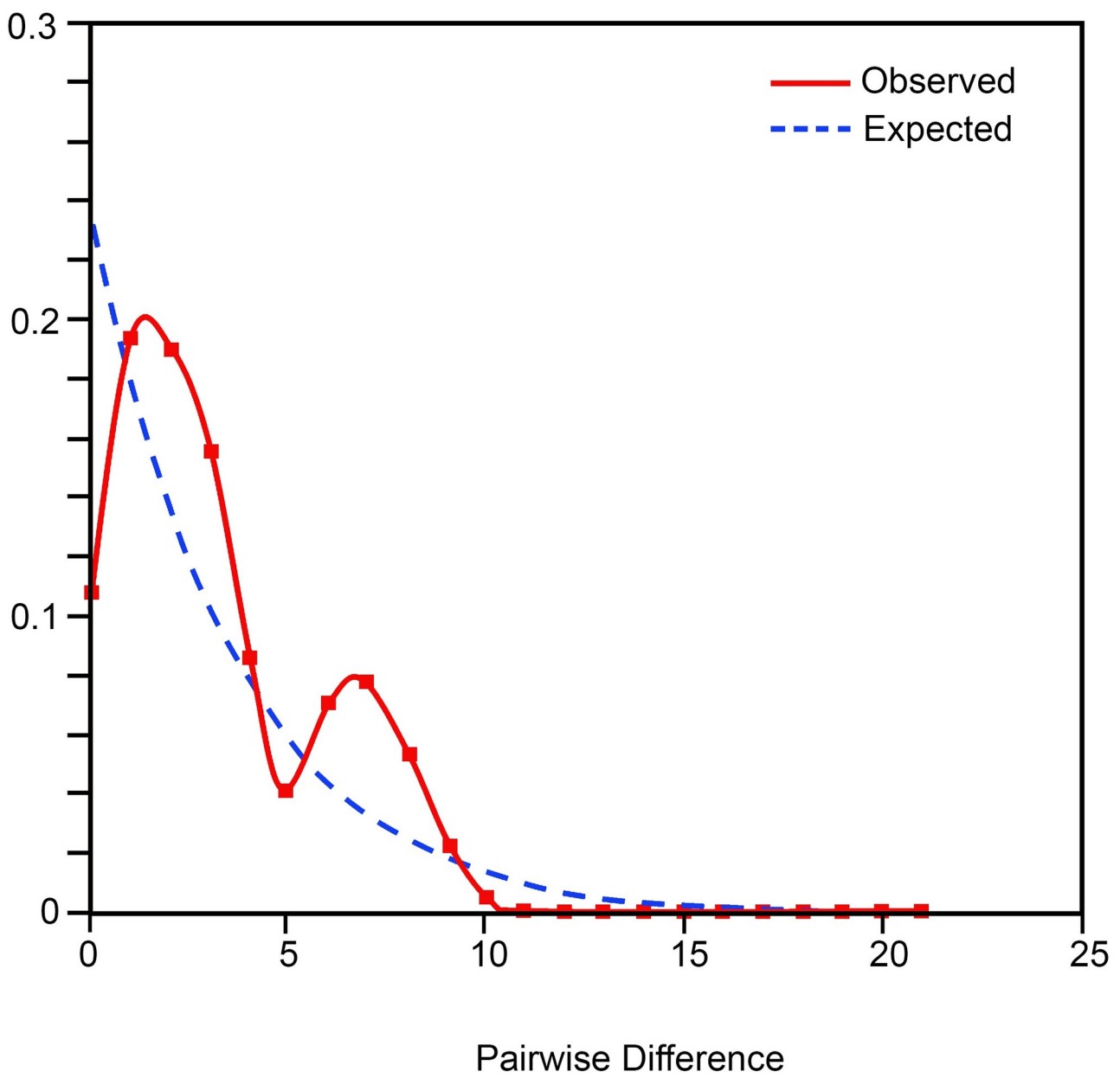

**Fig 5. Distribution curves of mismatch distribution for *cox1*+16S analysis of *Rhinocypha fenestrella* populations based on pairwise differences among haplotypes.** Note: The solid red line represents the actual observed distribution; the blue dashed line represents the expected distribution.

## Discussion

### Genetic diversity of Malaysian *Rhinocypha fenestrella*

Throughout the last five decades, the understanding of the ecology and evolution of odonates has increased dramatically. Recent advances in molecular techniques have inspired several phylogeographical studies of Odonata using genetic data [24,39,40]. As is known, most of the odonates have varying levels of dispersal abilities [41] that could influence the genetic diversity and phylogeographic structures of the populations. A less-mobile species may be expected to show some evidence of haplotype clustering according to geographic region [42]. Therefore, in

this study, we defined the intraspecific genetic diversity, phylogeographical patterns, and mitochondrial variations of the *Rhinocypha fenestrella* using *cox1* and 16S genes.

The variability of obtained sequences among all studied individuals for both markers ranged from 1.5–4.6% reflected in 28 variable nucleotide sites in 614 sequenced base pairs from *cox1* gene, and eight variable nucleotides in 534 base pairs from 16S gene. The base composition of both *cox1* and 16S gene sequences showed a significant A+T nucleotide bias which is consistent with insect mitochondrial genes [39,43,44].

In the present study, overall genetic diversity ($H_d$ = 0.8937; $P_i$ = 0.0028) of Malaysian *R. fenestrella* based on the concatenated *cox1*+16S sequences was slightly lower if compared to the study that did on the damselfly *Matrona basilaris* populations from mainland China [45]. The study reported higher values of haplotype diversity ($H_d$ = 0.9750) and nucleotide diversity ($P_i$ = 0.0049) by using *cox1* sequences. Nevertheless, the genetic diversity of *R. fenestrella* in Malaysia can be considered as high when compared to other damselflies species from the same sub-order *Psolodesmus mandarinus* (MacLahlan, 1870) from Taiwan that reported haplotype diversity as 0.43 and nucleotide diversity as 0.0017 based on *cox1*+16S sequences [46].

Among the sampling locations, Pahang population showed the highest haplotype numbers. Pahang, being a largest state in Malaysia, may have diverse geographic features such as rivers, mountains, and forests that probably contributed to the isolation of populations, hence promoting genetic divergence, and increasing the likelihood of different haplotypes. Geographical barriers known to restrict the movement of individuals between populations, reducing the gene flow. Consequently, limited gene flow allows for the accumulation of genetic differences between populations, leading to increased genetic diversity over time [47].

## Haplotype variation and distribution of *Rhinocypha fenestrella*

In this study, 26 haplotypes were revealed by *cox1*, 10 haplotypes by 16S and 32 haplotypes by concatenated sequences. The number of haplotypes detected in this study was higher compared to other similar studies by Lin et al. (2012) [46] that reported only 14 haplotypes variation in the damselfly population of *Psolodesmus mandarinus* from Taiwan. The high numbers of haplotypes are due to a high degree of *cox1* and 16S gene polymorphisms in *R. fenestrella* samples in Peninsular Malaysia. In fact, the polymorphism that occurred in Odonata were widely reported in other studies [48,49]. Whereas this study showed extremely low haplotype numbers when compared to study done by Jiang et al. (2023) [50] on the population of damselfly *Ischnura senegalensis* (Rambur, 1842) that revealed of 51 haplotype variations.

The haplotype AB2 appeared as the most dominant haplotype, while the limited geographical distribution of some of the haplotypes including those with singleton sites suggests the existence of genetic differentiation within the populations. In this study, haplotype AB1 was suggested as the common ancestor which appear as central haplotype producing a star-like radiation in the haplotype network, indicating the divergence of other haplotypes from its polymorphic sequence. This haplotype probably had eventually evolved over time into the numerous haplotypes (AB2 –AB15) to adapt to the habitat and demographic changes and consequently distributed across states in Peninsular Malaysia. Likewise, the most recent common ancestor for *R. fenestrella* in Malaysia may be derived from Negeri Sembilan as this state is constituted by high frequencies of the AB1 haplotypes.

Nevertheless, the network revealed no apparent geographical pattern, indicating a lack of genetic structuring across different populations in Peninsular Malaysia. The median-joining network revealed a close relationship among haplotypes, suggesting that *R. fenestrella* populations in Malaysia shared a recent history without long-term genetic isolation. Nevertheless, the ancestral haplotype AB1 of concatenated *cox1*+16S dataset was having a limited distribution

with restricted geographic division. This indicates that geographic barriers and climatic factors could also have little influence on the dispersion of this damselflies into different habitats in Malaysia besides their less ability to disperse due to weak flyers [51].

In mismatch distribution analysis, the peak curves indicated that the *R. fenestrella* populations had underwent an expansion process though the graph represented a bimodal characteristic which indicate no recent population expansion. The hypothesized that the bimodal pattern of mismatch distribution shown by the *R. fenestrella* populations could probably be due to the low migration rate of the samples within the studied locations. This aligned with the behaviour of the species that known as weak flyer and lack of dispersion capability [51].

## Genetic distance and differentiation

In our dataset, the highest genetic distance value based on *cox1* gene was 0.16%, while 16S gene was 0.75%. The data shows a higher genetic distance among Malaysian *R. fenestrella* as compared to the populations of *Rhinocypha taiwana* (Wang & Chang, 2013), *Rhinocypha uenoi* (Asahina, 1964), and *Rhinocypha drusilla* (Needham, 1930) from China where their highest genetic distances obtained based on *cox1* gene were 0.00%, 0.00% and 0.50%, respectively [52]. However, the genetic distance of Malaysian *R. fenestralla* was relatively lower if compared to other dragonfly populations of *Trithemis stictica* (Burmeister, 1839) in Namibia [9] and *Nannophya pygmaea* (Rambur, 1842) in Malaysia [53] that presented the genetic distance of *cox1* gene of up to 9.00% and 12.00%, respectively.

Based on AMOVA analysis, the major variation observed in this study was among the populations, indicating that samples have differentiated into separate genetic pools that could lead to genetic fragmentation, hence, genetic variation has led to a high level of differentiation among populations. Additionally, $N_m<1$ indicates of insufficient gene flow between populations leading to genetic differentiation and genetic isolation. *Rhinocypha fenestrella* is known as a weak flyer [51,54] and is unlikely to migrate over large-scale regions, which likely contributes to its overall high genetic differentiation to happen among the populations. Therefore, it is not surprising that genetic differentiation happened in the species *R. fenestrella* in which their populations were separated up to more than 500 km. In fact, the migratory behaviour of odonate could homogenize genetic differentiation among populations by the exchange of individuals and genes among populations with high genetic differentiation as observed in the population of migratory dragonflies *Libellula quadrimaculata* (Linnaeus, 1758) [39] and *Pantala flavescens* (Fabricius, 1798) [55] respectively, which are known as strong fliers than the damselflies.

Dispersal ability and long-distance migration are the most important factors contributing to a high level of gene flow and consequent slowing or limiting of geographic differentiation [56,57]. A less-mobile species may be expected to show some evidence of haplotype clustering according to geographic region due to less ability to disperse and therefore resulted in genetic isolation by distance. In this study, calculated $F_{st}$ value shows high genetic differentiation in the populations of *R. fenestrella* in Malaysia. This high value is consequently supported by the low rate of gene flow found amongst the *R. fenestrella* populations in Malaysia as well as between the different sites.

When all eight populations were regarded as a whole, Tajima's D statistic was statistically not significant. The results showed that these populations were in a stable state with no demographic expansion and no recent bottleneck. Whereas, in the Selangor population, Fu's $F_S$ statistic values were negative with *p*-values being significant ($p < 0.02$) which shows this population had experienced a recent population expansion. Our results suggested that *R. fenestrella* population in Selangor could successfully colonize and adapt to new habitats and were able to disperse randomly and exchange genes with local populations. Maintaining genetic

diversity is noteworthy as it related to population viability [58–60] and also to the transformative potential of a species to react to the environmental changes [61,62].

Overall, this study found strong evidence for intraspecific patterns of haplotype variation among populations of *R. fenestrella* in Malaysia though the weak dispersal abilities of the species. The study also revealed high genetic differentiation within the populations and a low rate of gene flow among the geographically difference populations of Malaysian *R. fenestrella*. Moreover, a high haplotype number was observed in *R. fenestrella* population, indicates the existence of genetic isolation within the populations of the sampling sites.

Although many phylogeographical mechanisms have been proposed using damselflies as model organisms, more detailed sampling and a larger variety of ecological investigations are required to promote better understanding. Therefore, this study contributed new insight for an advanced understanding of the evolution and phylogeography of damselfly. Nevertheless, additional sequence data from other regions outside of Malaysia (e.g.: Thailand, Burma, Laos, Vietnam, and southern China), which is considered a gap of knowledge due to the lack of genetic data on this particular species, may prove useful in detecting phylogenetic relationships and phylogeography patterns, and revealing common ancestors of these populations between the other regions and continents.

A comprehensive understanding of genetic diversity is crucial for effective biodiversity conservation. The findings from this study may contribute to broader initiatives aimed at preserving biodiversity in the region especially in Peninsular Malaysia. Different populations may have unique ecological adaptations, and management efforts can consider these variations for more effective conservation. Moreover, monitoring genetic diversity over time can help assess the impact of environmental changes on the damselfly population. The information obtained from this study could have implications for understanding the species' ability to adapt to changing environments.

## Supporting information

**S1 Table. Percentage (%) of uncorrected "*p*" distance matrix among the 26 representative *cox1* haplotypes of *Rhinocypha fenestrella* in Malaysia.**
(PDF)

**S2 Table. Percentage (%) of uncorrected "*p*" distance matrix among the 10 representative 16S rRNA haplotypes of *Rhinocypha fenestrella* in Malaysia.**
(PDF)

**S3 Table. Percentage (%) of uncorrected "*p*" distance matrix among the 32 representatives for the concatenated *cox1*+16S haplotypes of *Rhinocypha fenestrella* in Malaysia.**
(PDF)

## Acknowledgments

The authors would like to thank Mr. Mohaiyidin Mohamed and Mr. Mohd Fauzi Abd Hamid for their assistance in the fieldwork.

## Author Contributions

**Conceptualization:** Mamat Noorhidayah, Norma-Rashid Yusoff.

**Data curation:** Mamat Noorhidayah.

**Formal analysis:** Mamat Noorhidayah, Noor Azrizal-Wahid, Van Lun Low.

**Funding acquisition:** Norma-Rashid Yusoff.

**Investigation:** Mamat Noorhidayah, Norma-Rashid Yusoff.

**Methodology:** Mamat Noorhidayah, Norma-Rashid Yusoff.

**Project administration:** Norma-Rashid Yusoff.

**Supervision:** Norma-Rashid Yusoff.

**Visualization:** Mamat Noorhidayah, Noor Azrizal-Wahid, Van Lun Low.

**Writing – original draft:** Mamat Noorhidayah, Noor Azrizal-Wahid.

**Writing – review & editing:** Noor Azrizal-Wahid, Van Lun Low.

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
