## [Decision Letter · Decision Letter 0]

25 Sep 2023

PONE-D-23-28970Genetic diversity and phylogeographic patterns of the peacock jewel-damselfly, Rhinocypha fenestrella (Odonata: Chlorocyphidae)PLOS ONE

Dear Dr. Abdul Wahid,

Thank you for submitting your manuscript to PLOS ONE. After careful consideration, we feel that it has merit but does not fully meet PLOS ONE’s publication criteria as it currently stands. Therefore, we invite you to submit a revised version of the manuscript that addresses the points raised during the review process.

We look forward to receiving your revised manuscript.

Kind regards,

Tzen-Yuh Chiang

Academic Editor

PLOS ONE

Journal Requirements:

https://www.nature.com/articles/s41598-019-45123-0?code=09df2bbc-e733-4183-ad07-99f2eea3b745&error=cookies_not_supported

In your revision ensure you cite all your sources (including your own works), and quote or rephrase any duplicated text outside the methods section. Further consideration is dependent on these concerns being addressed.

4. Thank you for stating the following financial disclosure: "Author who received the funding award= Norma-Rashid Yusoff;  

Grant numbers awarded to the author= 1;

The full name of funder= The Universiti Malaya Research Grant (UMRG) PG065 – 2015 and ST061-2022;

URL of each funder website= https://umhvl.um.edu.my/grants-and-projects-awarded;

Role of the funder: Conceptualization, Investigation, Methodology, Project administration, Supervision"

Please state what role the funders took in the study.  If the funders had no role, please state: The funders had no role in study design, data collection and analysis, decision to publish, or preparation of the manuscript.

5. Thank you for stating the following in the Acknowledgments Section of your manuscript: "This study was supported by a research grant from the University of Malaya, PG065 – 2015 and ST061-2022. The authors would like to thank Mr. Mohaiyidin Mohamed and Mr. Mohd Fauzi Abd Hamid for their assistance in the fieldwork".

Please remove any funding-related text from the manuscript and let us know how you would like to update your Funding Statement. Currently, your Funding Statement reads as follows: "Author who received the funding award= Norma-Rashid Yusoff;  

Grant numbers awarded to the author= 1;

The full name of funder= The Universiti Malaya Research Grant (UMRG) PG065 – 2015 and ST061-2022;

URL of each funder website= https://umhvl.um.edu.my/grants-and-projects-awarded;

Role of the funder: Conceptualization, Investigation, Methodology, Project administration, Supervision".

7. We note that Figure 1 in your submission contain [map/satellite] images which may be copyrighted. All PLOS content is published under the Creative Commons Attribution License (CC BY 4.0), which means that the manuscript, images, and Supporting Information files will be freely available online, and any third party is permitted to access, download, copy, distribute, and use these materials in any way, even commercially, with proper attribution. For these reasons, we cannot publish previously copyrighted maps or satellite images created using proprietary data, such as Google software (Google Maps, Street View, and Earth). For more information, see our copyright guidelines: http://journals.plos.org/plosone/s/licenses-and-copyright.

8. Please include a copy of Tables 1, 2, 3, 4 and 5 which you refer to in your text on pages 3, 6, 7, and 8.

10. Please upload a copy of Supporting Information Figure/Table/etc. S2 Table B, which you refer to in your text on page 21.

Reviewers' comments:

Reviewer's Responses to Questions

**Comments to the Author**

1. Is the manuscript technically sound, and do the data support the conclusions?

Reviewer #1: Partly

Reviewer #2: Partly

2. Has the statistical analysis been performed appropriately and rigorously? 

Reviewer #1: Yes

Reviewer #2: No

3. Have the authors made all data underlying the findings in their manuscript fully available?

Reviewer #1: Yes

Reviewer #2: Yes

4. Is the manuscript presented in an intelligible fashion and written in standard English?

Reviewer #1: No

Reviewer #2: No

5. Review Comments to the Author

Reviewer #1: The number of populations and sequences seem appropriate, and the analyses are acceptable. However, MANOVA and Mantel tests should also be performed. There are no conclusions about the divergence pattern. The discussion is somewhat chaotic and not sound, a lot more should be said on one hand, and several repeated statements should be eliminated, also some contradictory opinions. Please see my comments in the text. I am not a native speaker, but English should be corrected - at some parts it is simply impossible to understand the text. I have made some suggestions, but these are only suggestions, and much more should be corrected. Also some strange terminology can be found, like "uncorrected genetic diversity".

Reviewer #2: In the present study the genetic diversity and phylogeography of the Rhinocypha fenestrella were studied. The authors used the cytochrome C oxidase I and 16S rRNA gene regions from 147 individuals representing eight populations in Malaysia. The research is interesting, but to make it suitable for printing, additions and corrections are needed, however, I believe that not all of them are possible. The work has flaws that are difficult to correct. Eight places are not enough to draw phylogeographic conclusions ( I am of course aware of all the problems that may arise during field work), however, the number of individuals per population is sufficient to conduct basic population analyses. The second problem can be removed - the authors used only fragments of mtDNA, so they are actually analyzing the phylogeny of the maternal line. Perhaps it would be worth adding markers for nuclear genes, histone, 18S, 28S, they should be sequenced relatively quickly, and they would undoubtedly significantly increase the value of the work. There is also a problem with the low observed diversity. In fact, the work describes small variability within undoubtedly one species based only on mtDNA and it is not possible to draw general conclusions on this basis. Taking the above into account, it may be worth considering publishing the results in less reputable journals than PlosOne

Other comments:

- authors should also use BOLD Systems for sequences identification

- MUSCLE is better than CLUSTAL X

- the authors should perform a full phylogenetic analysis using maximum likelihood and Bayesian approaches (maybe some mOTUs will be showed) with comparison to closely related species and detailed analysis. As I have already mentioned, small genetic diversity and a small number of sites may make such analyzes much more difficult

- the sequence length is given unnecessarily in two different places

- number of sequences and the GB numbers are results not methods

- comparing genetic distances and Fst with geographic distances between populations should enrich the results

- sites should be marked on map, this same for haplotypes, or maybe mOTUs?

- based on their data, the authors drew too far-reaching conclusions about the evolutionary history of the species under study

I am not native spiker, but in my opinion some linguistic corrections should be made

6. PLOS authors have the option to publish the peer review history of their article (what does this mean?). If published, this will include your full peer review and any attached files.

Reviewer #1: **Yes: **Andrzej Falniowski

Reviewer #2: No

---

## [Author Response · Author response to Decision Letter 0]

8 Oct 2023

The detail responses to the editors and reviewers have been provided as attached file (see .doc file with title "Response to Reviewers". Thank you.

---

## [Decision Letter · Decision Letter 1]

12 Dec 2023

PONE-D-23-28970R1Genetic diversity and phylogeographic patterns of the peacock jewel-damselfly, Rhinocypha fenestrella (Rambur, 1842)PLOS ONE

Dear Dr. Abdul Wahid,

Thank you for submitting your manuscript to PLOS ONE. After careful consideration, we feel that it has merit but does not fully meet PLOS ONE’s publication criteria as it currently stands. Therefore, we invite you to submit a revised version of the manuscript that addresses the points raised during the review process.

We look forward to receiving your revised manuscript.

Kind regards,

Tzen-Yuh Chiang

Academic Editor

PLOS ONE

Reviewers' comments:

Reviewer's Responses to Questions

**Comments to the Author**

1. If the authors have adequately addressed your comments raised in a previous round of review and you feel that this manuscript is now acceptable for publication, you may indicate that here to bypass the “Comments to the Author” section, enter your conflict of interest statement in the “Confidential to Editor” section, and submit your "Accept" recommendation.

Reviewer #3: (No Response)

Reviewer #4: (No Response)

Reviewer #5: All comments have been addressed

Reviewer #6: All comments have been addressed

Reviewer #7: All comments have been addressed

2. Is the manuscript technically sound, and do the data support the conclusions?

Reviewer #3: Partly

Reviewer #4: Yes

Reviewer #5: Partly

Reviewer #6: Yes

Reviewer #7: Partly

3. Has the statistical analysis been performed appropriately and rigorously? 

Reviewer #3: Yes

Reviewer #4: Yes

Reviewer #5: No

Reviewer #6: Yes

Reviewer #7: Yes

4. Have the authors made all data underlying the findings in their manuscript fully available?

Reviewer #3: Yes

Reviewer #4: Yes

Reviewer #5: Yes

Reviewer #6: Yes

Reviewer #7: Yes

5. Is the manuscript presented in an intelligible fashion and written in standard English?

Reviewer #3: No

Reviewer #4: Yes

Reviewer #5: Yes

Reviewer #6: Yes

Reviewer #7: Yes

6. Review Comments to the Author

Reviewer #3: The manuscript contains very useful information regarding the genetic population structure of the damselfly Rhinocypha fenestrella in Malaysia. However, there are several issues that should be addressed by the authors before the manuscript is accepted for publication

1. Include the country where the study was conducted in the title

2. The abstract does not contain a statement of the problem

3. Include in the abstract the implications of these results

4. Page 2, line 16, you are talking about dragonflies - the title is about damselflies - note that damselflies are completely different from dragonflies

5. The introduction is very short – the biology, particularly life history of the studied species is not explained in the introduction

6. Page 4, line 9, what was the concentration of ethanol used to preserve the samples

7. Page 6, line 9 – 12, focus on the p-value. Check whether there Fst values are significant – forget about the classification

8. Correct grammatical errors in page 6, lines 23 – 24

9. Table 2, the measured Hd for Johor, Perak and Selangor based on 16S is zero – this could be attributed to low sample size. Example, looking at Table 3, it appears, you only sampled 11 individuals from Selangor. Do you this this number is enough to give you meaningful comparisons of the indices of genetic diversity between populations?

10. Subtitles in page 6 line 22 and page 8 line 2 have the same meaning

11. State whether the Fst values in Table 4 are significant. If they are not significantly different from zero – then your explanations in page 12, lines 2 – 7 do not make any sense

12. Page 12 line 8, was the Fus’ values significant?

13. Page 12, line 11 to 12, correct grammatical errors

14. Page 12, line 12 to 15, look at the values you are calling high and low – (0.0100 and 0.0010). Are they significantly different from zero? If they are not significantly different from zero, then it suggests that these populations are genetically homogenous. Please – revisit the interpretation of your results

15. Correct grammatical errors in page 12 line 18

16. Page 13, line 21, comparing diversity indices from different markers does not make sense

17. Page 14, line 5, high compared to what?

18. Page 14, line 11, you are comparing different and unrelated species

19. Page 14, line 14 – 15, check the interpretation – it is not correct. Having singletons in a population does not necessarily mean the population is isolated from the others

20. Page 15, lines 7 – 15 requires serious revisions – interpretations given here are not correct

21. Page 16, lines 7 - 8, revisit your interpretations

22. Page 16, lines 12 and 13, interpretations of Tajima values should be rechecked

23. Interpretations of results in page 16, lines 23 – 25 are conflicting each other

24. Phylogenetic analysis of Rhinocypha fenestrella is missing in the entire document. Revise the title or do phylogenetic analysis of the studied species. Note - This was advised by the previous reviewer. The author just ignored this comment but his defense of this omission lacks coherence and fails to address the critical need for this analysis. Further attention is warranted to rectify this deficiency and enhance the study's comprehensiveness.

25. What are the implications of this study on the management of aquatic ecosystems?

Reviewer #4: Dear Editor in chief,

I have reviewed the manuscript; several flaws must be must properly be taken care of.

1. Why did the author use a concatenated sequence for their analysis? All the discussion and interpretation of the results were based on concatenated sequences.

2. In Table 2, The number of sequences (N) should also be used in each sampling site for each marker.

3. In Table 3, edit (n) the number of sequences. Also, they should mention the number of haplotypes for each site on the right side of the table.

4. Please discuss the possible reason for the highest number of haplotypes in Pahang.

5. Please discuss why the diversity in Selangor, with the lowest number of haplotypes, was higher than in Kedah.

6. Why didn't the authors consider sequences from other countries? They should include other sequences from neighboring countries to show the gene flow and genetic diversity between Malaysia and other countries.

7. They should also use a phylogenetic tree to illustrate their result better.

Reviewer #5: This manuscript aimed to explore the genetic diversity and observed phylogeogrphic pattern of Rhinocypha fenestrella using two different markers along Malay peninsula. Frankly, the revised manuscript appears to be much better. However, I still find some points can be improved and some analysis need to be added into this MS, and I believe that it will make this MS more attractive and informative. I have added some comments into the file, please see the attached. Also some major concerned are listed below:

1. I think instead of showing only haplotype network (Figure 2-4), It must be nice if authors can add the map showing haplotype distribution in to each figure. It will be easy to see clearly about the haplotype diversity and promotion in each population.

2. As I found author responded to one of reviewer that this paper aim to emphasize on genetic diversity and population structure analysis. I think AMOVA analysis is really needed to be done. Also mismatch distribution or skyline plot should be calculated to see demographic history of this species in Malaysia too.

3. I don't think DNAsp is suitable to use for Fst calculations as it did not provide significant level. Please rerun Fat in Alrequin or GenALEx.

4. Author mentioned low gene flow, but table 4 show very low genetic differentiation between the population. I think it is incongruous. Please reanalyze.

Good luck

Reviewer #6: REF. Manuscript Number: PONE-D-23-28970R1

The manuscript entitled “Genetic diversity and phylogeographic patterns of the peacock jewel-damselfly, Rhinocypha fenestrella (Rambur, 1842)” by Mamat-Noorhidayah et al. demonstrated the study on population structure and haplotype diversity of the damselfly, R. fenestrella. The revised manuscript has been made accordingly to the editor and reviewer’s comments. The current form of the paper is well written with acceptable English present. The rationale, scientific logic, methods, and analysis of data are appropriate.

Reviewer #7: The findings about the intraspecific genetic diversity, phylogeographic trend, and the ancestral haplotype of the peacock jewel-damselfly are remarkable. However, the authors need to address the following concerns:

1. Deviation from a standard neutral model is restrictive per gene and more informative when each gene is analyzed separately. Concatenated data may give a false impression of Tajima’s D, Fu’s and Li, Fu’s or Fu and Li’s D*, Fu and Li’s F*, and Fu’s FS statistic.

Abstract

2. Page 1, Line 24-25 needs to be rephrased for clarity.

Methodology

3. Confirm the 300 sec on pcr protocol.

4. …..The aligned cox1 and 16S sequences at first were analysed respectively…… were analysed separately….

5. The FST and Nm pairwise values were calculated to access genetic differentiation and gene flow among the R. fenestrella population and were calculated using DNASP®… Rephrase the sentence for clarity.

Discussion

1. …..The obtained sequences for cox1 and 16S genes from eight populations ranged from 1.5……….. Rephrase the sentence for clarity.

2. In this study, the number of haplotypes detected was considered high with 26 haplotypes were revealed by cox1 and 10 haplotypes by 16S. Rephrase the sentence for clarity.

3. Page 14 Line 6-7 The high number is due to and not “may indicate” ..... It is an indicative of high diversity in species.

4. Page 14 Line 14 …..with singleton sites…….. also include Reference attributing presence of singleton sites to current genetic differentiation.

5. …..to star-like radiation in……This is not an indicator of common ancestry…….Is a common ancestry due divergence of haplotype from its polymorphic sequence.

6. ….Likewise, according to location20 based, the most recent common ancestor for R. fenestrella in Malaysia may be derived from Negeri Sembilan as this state is constituted by high frequencies of the common ancestor AB1 haplotypes. Rephrase the sentence to avoid repetition.

7. ….The data shows a higher genetic distance among Malaysian R. fenestrella as compared to the populations of Rhinocypha taiwana (Wang & Chang, 2013), Rhinocypha uenoi (Asahina, 1964), and Rhinocypha drusilla (Needham, 1930) from China where the highest genetic distances based on cox1 gene were 0.00%, 0.00% and 0.50%, respectively……….0.00% is not high

8. ……cox1 gene of up to 9.00% and 12.00%, respectively……………..

9. …..Zygopterans are known as weaker fliers since many species do not diffusing outside several kilometres……rewrite for clarity.

7. PLOS authors have the option to publish the peer review history of their article (what does this mean?). If published, this will include your full peer review and any attached files.

Reviewer #3: No

Reviewer #4: No

Reviewer #5: No

Reviewer #6: No

Reviewer #7: **Yes: **Kevin O. Ochwedo

---

## [Author Response · Author response to Decision Letter 1]

29 Feb 2024

The authors would like to thank to all reviewers for acknowledge our great efforts on preparing this manuscript. We highly appreciated all the given comments to improvise the writing and content of the manuscript in which we believe that our revised manuscript is now have met the standard requirements to be published in PLOS ONE journal where the knowledge and information rehashed and discussed in this manuscript will be of great value to readers and researchers alike. We have addressed all the comments and the responses to the comments point to point are available in attached document.

---

## [Decision Letter · Decision Letter 2]

14 Mar 2024

Genetic diversity and phylogeographic patterns of the peacock jewel-damselfly, Rhinocypha fenestrella (Rambur, 1842)

PONE-D-23-28970R2

Dear Dr. Abdul Wahid,

We’re pleased to inform you that your manuscript has been judged scientifically suitable for publication and will be formally accepted for publication once it meets all outstanding technical requirements.

Kind regards,

Tzen-Yuh Chiang

Academic Editor

PLOS ONE

Additional Editor Comments (optional):

Reviewers' comments:

Reviewer's Responses to Questions

**Comments to the Author**

1. If the authors have adequately addressed your comments raised in a previous round of review and you feel that this manuscript is now acceptable for publication, you may indicate that here to bypass the “Comments to the Author” section, enter your conflict of interest statement in the “Confidential to Editor” section, and submit your "Accept" recommendation.

Reviewer #4: All comments have been addressed

Reviewer #6: All comments have been addressed

Reviewer #7: All comments have been addressed

2. Is the manuscript technically sound, and do the data support the conclusions?

Reviewer #4: Yes

Reviewer #6: Yes

Reviewer #7: Yes

3. Has the statistical analysis been performed appropriately and rigorously? 

Reviewer #4: Yes

Reviewer #6: Yes

Reviewer #7: Yes

4. Have the authors made all data underlying the findings in their manuscript fully available?

Reviewer #4: Yes

Reviewer #6: Yes

Reviewer #7: Yes

5. Is the manuscript presented in an intelligible fashion and written in standard English?

Reviewer #4: Yes

Reviewer #6: Yes

Reviewer #7: Yes

6. Review Comments to the Author

Reviewer #4: As they did not include the sequences of other countries, it would be better to mention the country name in the title.

Reviewer #6: The authors have response to all comments and suggestions. The paper has been improved and it current form is accepted for publication.

Reviewer #7: The authors have adequately addressed all the comments raised. The current form of the manuscript, "Genetic diversity and phylogeographic patterns of the peacock jewel-damselfly, Rhinocypha fenestrella (Rambur, 1842)," is well written, and all the previously noted typos have been corrected.

7. PLOS authors have the option to publish the peer review history of their article (what does this mean?). If published, this will include your full peer review and any attached files.

Reviewer #4: **Yes: **Elham Kazemirad

Reviewer #6: No

Reviewer #7: **Yes: **Kevin Omondi Ochwedo

---

## [Editor Report · Acceptance letter]

20 Mar 2024

PONE-D-23-28970R2 

PLOS ONE

Dear Dr. Azrizal-Wahid, 

I'm pleased to inform you that your manuscript has been deemed suitable for publication in PLOS ONE. Congratulations! Your manuscript is now being handed over to our production team.

Kind regards, 

on behalf of

Dr. Tzen-Yuh Chiang 

Academic Editor

PLOS ONE